# Tumor Suppressor Properties of Small C-Terminal Domain Phosphatases in Clear Cell Renal Cell Carcinoma

**DOI:** 10.3390/ijms241612986

**Published:** 2023-08-19

**Authors:** George S. Krasnov, Grigory A. Puzanov, Erdem B. Dashinimaev, Khava S. Vishnyakova, Tatiana T. Kondratieva, Yegor S. Chegodaev, Anton Y. Postnov, Vera N. Senchenko, Yegor E. Yegorov

**Affiliations:** 1Engelhardt Institute of Molecular Biology, Russian Academy of Sciences, 119991 Moscow, Russia; inrainbow@mail.ru (G.A.P.); khava58@mail.ru (K.S.V.); egozavr-ch@mail.ru (Y.S.C.); verasenchenko@gmail.com (V.N.S.); 2Center for Precision Genome Editing and Genetic Technologies for Biomedicine, Engelhardt Institute of Molecular Biology, Russian Academy of Sciences, 119991 Moscow, Russia; 3Center for Precision Genome Editing and Genetic Technologies for Biomedicine, Pirogov Russian National Research Medical University, Ostrovitianov Street, 117997 Moscow, Russia; dashinimaev_eb@rsmu.ru; 4Research Institute of Clinical Oncology, Blokhin National Medical Research Center of Oncology of the Ministry of Health, 115478 Moscow, Russia; ttkondr@gmail.com; 5Eurasian Federation of Oncology, 125080 Moscow, Russia; 6Laboratory of Cellular and Molecular Pathology of Cardiovascular System, Federal State Budgetary Scientific Institution “Petrovsky National Research Centre of Surgery”, 119991 Moscow, Russia; anton-5@mail.ru

**Keywords:** renal cancer, SCP/CTDSP phosphatases, tumor suppressor activity, TCGA, survival analysis

## Abstract

Clear cell renal cell carcinoma (ccRCC) accounts for 80–90% of kidney cancers worldwide. Small C-terminal domain phosphatases CTDSP1, CTDSP2, and CTDSPL (also known as SCP1, 2, 3) are involved in the regulation of several important pathways associated with carcinogenesis. In various cancer types, these phosphatases may demonstrate either antitumor or oncogenic activity. Tumor-suppressive activity of these phosphatases in kidney cancer has been shown previously, but in general case, the antitumor activity may be dependent on the choice of cell line. In the present work, transfection of the Caki-1 cell line (ccRCC morphologic phenotype) with expression constructs containing the coding regions of these genes resulted in inhibition of cell growth in vitro in the case of *CTDSP1* (*p* < 0.001) and *CTDSPL* (*p* < 0.05) but not *CTDSP2*. The analysis of The Cancer Genome Atlas (TCGA) data showed differential expression of some of *CTDSP* genes and of their target, *RB1*. These results were confirmed by quantitative RT-PCR using an independent sample of primary ccRCC tumors (*n* = 52). We observed *CTDSPL* downregulation and found a positive correlation of expression for two gene pairs: *CTDSP1* and *CTDSP2* (*r_s_* = 0.76; *p* < 0.001) and *CTDSPL* and *RB1* (*r_s_* = 0.38; *p* < 0.05). Survival analysis based on TCGA data demonstrated a strong association of lower expression of *CTDSP1*, *CTDSP2*, *CTDSPL*, and *RB1* with poor survival of ccRCC patients (*p* < 0.001). In addition, according to TCGA, *CTDSP1*, *CTDSP2*, and *RB1* were differently expressed in two subtypes of ccRCC—ccA and ccB, characterized by different survival rates. These results confirm that *CTDSP1* and *CTDSPL* have tumor suppressor properties in ccRCC and reflect their association with the more aggressive ccRCC phenotype.

## 1. Introduction

Clear cell renal cell carcinoma (ccRCC) represents the most aggressive histological form of kidney cancer and accounts for 80–90% of cases of renal carcinomas [1]. ccRCC is often diagnosed at late stages, and the survival rate when metastases are present is extremely low. The outcomes of ccRCC are the worst among all urogenital tumors. ccRCC represents about 3% of all cancers occurring with the highest incidence in Western countries. ccRCC is the 13th leading cause of cancer deaths worldwide [2].

In general, ccRCC is well circumscribed and the capsule is usually absent. Loss of chromosome 3p and mutation of the von Hippel-Lindau (*VHL*) gene located at chromosome 3p25 are common. Loss of function of the von Hippel-Lindau protein contributes to tumor initiation, progression, and metastasis. Additional ccRCC tumor suppressor genes (*UTX*, *JARID1C*, *SETD2*, *PBRM1*, and *BAP1*) are located at the 3p locus [3].

The sub-family of small C-terminal domain phosphatases (SCP) contains four members—CTDSP1 (also referred to as SCP1), CTDSP2 (SCP2), CTDSPL (SCP3), and CTDSPL2 (SCP4). These enzymes play important roles in numerous cellular processes [4], including cell proliferation and apoptosis [5,6]; differentiation [7,8], which is often dysregulated in different types of cancer, as well as axonal regeneration [9]; and suppression of neuronal-specific genes in non-neuronal tissues [10]. All CTDSP proteins include a phosphatase domain, which is homologous to the catalytic domain of F-cell production 1 (FCP1) protein, an essential RNA polymerase II Ser/Thr phosphatase [11]. The phosphatase domains of CTDSP1, 2, and L share high homology with each other (83–85%), whereas CTDSPL2 demonstrates significantly less homology (only 65%). 

In this work, we focused on the study of phosphatases CTDSP1, 2, and L. Their involvement in carcinogenesis was demonstrated for various cancers. Depending on tumor type, these phosphatases may exhibit either oncogenic or tumor-suppressive activity. This is mainly due to the diversity of their targets, which include Rb1 [12], Twist1 [13], AKT [14], SMAD2 and 3 [8], SNAI1 [15], REST [10], p21(Cip1/Waf1) [6], c-Myc [5], and PML [16]. The most known and best studied CTDSP1/2/L target is the retinoblastoma protein (Rb1), a potent tumor suppressor.

Earlier, Yu-Ching Lin et al. showed the tumor-suppressive properties of SCP phosphatases in ccRCC, which are mediated by the stabilization of the PML protein [16]. Although the present work partially overlaps with the study of Lin et al., we used a different cell line model and obtained different results: tumor-suppressing activity was found only for CTDSP1 and CTDSPL. The cancer significance of genes can be quite different depending on the cell line used for evaluation. Different cell lines, even belonging to the same tumor type, may carry a different spectrum of driver mutations/deletions and thus involve different pathways of carcinogenesis.

Earlier it was shown that CTDSP1 inhibits breast cancer cell migration and invasion [13] and suppresses tumor properties of liver carcinoma [5], osteosarcoma [6], and uveal melanoma [17]. CTDSP1 also negatively regulates angiogenesis [14] and increases irinotecan sensitivity of colorectal cancer through the stabilization of topoisomerase I [18]. The *CTDSPL* gene is located in 3p22, a region prone to frequent mutations, deletions, and promoter methylations in many types of tumors, including lung, cervical, kidney, breast, and ovarian cancers [19,20]. Recently, we demonstrated that CTDSP1, 2, and L inhibit the growth of A549 cells, a non-small-cell lung cancer cell line [21]. The purpose of this paper was to extend these findings with regard to ccRCC. However, the results obtained on ccRCC were not so uniform as for lung cancer. This indicates a high heterogeneity of ccRCC.

The pro-oncogenic properties of CTDSP1, 2, and L phosphatases are also diverse. It was shown that CTDSP1 stabilizes the SNAI1 transcription factor, a key regulator of epithelial-mesenchymal transition, and thus promotes gastric cancer cell migration [15]. Moreover, it was demonstrated that CTDSP1 contributes to the increased migration of neuroglioma cells [22]. The activation of CTDSPL may be an early event in avian leukosis virus-induced carcinogenesis of B-cell lymphomas [23].

Although over the past decades, some success has been achieved in the treatment of kidney cancer, ccRCC is still characterized by high mortality, which determines the importance of the search for new mechanisms of carcinogenesis and the identification of tumor suppressors that are relevant for ccRCC.

## 2. Results 

### 2.1. Analysis of TCGA Omics Data for SCP Subfamily and RB1 in ccRCC 

Using our previously developed CrossHub tool, we analyzed RNA sequencing and methylation profiling data for the members of the SCP family in ccRCC. Among the three members of the SCP family and their primary target, *RB1*, only *CTDSPL* demonstrated significant downregulation in ccRCC in most samples (three-fold on average; see Appendix A). *RB1* and *CTDSP1* were characterized by weak overexpression (1.3-fold). *CTDSPL* and *CTDSP2* were significantly co-expressed (*r_s_* = 0.46; Appendix A). 

To identify the possible mechanisms of *CTDSPL* downregulation, we examined whether promoter CpG islands were hypermethylated in ccRCC. However, there were no significant changes in promoter region methylation levels for any of the four studied genes. We only observed hypermethylation of some intronic CpG sites that were distant from the promoter regions (Appendix A). The most pronounced intronic hypermethylation was observed for *CTDSPL*, where it was negatively correlated with gene expression (*r_s_* values range from −0.4 to −0.64, *p* < 10^−14^); we also observed hypermethylation of intronic sites of *CTDSP2* (*r_s_* values range from −0.24 to −0.32, *p* < 10^−6^) and *RB1* (no correlations with expression). According to ENCODE data on chromatin segmentation (ChromHMM/Segway algorithms), these sites may be located in the enhancer regions. Finally, the analysis of mutations in *CTDSP1/2/L* and *RB1* genes showed that somatic mutations in these genes are not a frequent event in ccRCC. 

Next, we searched for microRNAs (miRNAs) that were strongly anti-co-expressed with any of the four studied genes and had binding sites in 5′ UTRs, as such miRNAs may have regulatory potential. The most interesting results were those obtained for the *CTDSPL* gene, which demonstrated both anti-correlation with mir-18a, mir-181 a/b/d, mir-34a, and the presence of binding sites for these miRNAs predicted simultaneously by several algorithms or databases (TargetScan, DIANA microT, and miRTarBase; Appendix A). We also noticed mir-183 and mir-182 for *CTDSP1*; mir-15a for *CTDSP2*; and mir-106a, mir-26a, and some others for *RB1*. In ccRCC, we did not observe any common regulatory miRNAs with pronounced anti-co-expression for all three phosphatases, as it was a case for lung cancer [21].

### 2.2. Expression Analysis of SCP Subfamily Genes and RB1 in ccRCC Using RT-qPCR

The results of the quantitative expression analysis for *CTDSP1/2/L* and *RB1* in 52 primary paired samples (ccRCC tumor tissue and matched normal tissue) are shown in Figure 1 and Table 1. The distribution of the relative expression levels of four genes by the ccRCC stage is presented in Figure 1: I stage (*n* = 22), II stage (*n* = 13), and III stage (*n* = 17). The mRNA levels of *CTDSP1* and *CTDSP2* were slightly and infrequently increased. The difference in *CTDSP1* expression levels between stages I and II was statistically significant (*p* < 0.05). In contrast to *CTDSP1* and *CTDSP2*, *CTDSPL* expression showed a noticeable decrease in 50% of samples, which was more pronounced at stage III than at stage II (*p* < 0.05, Mann-Whitney U-test). The mRNA level of the retinoblastoma gene *RB1* was noticeably increased in 50% of cases, and the difference in its expression levels between the II and III stages was statistically significant (*p* < 0.001). No significant differences in the expression of four studied genes were found between male and female patients, or between tumor samples with and without metastases. 

A correlation analysis was carried out for *CTDSP1/2/L* and *RB1* expression (Appendix A). The results showed that the expression levels of *CTDSP1* and *CTDSP2* were highly correlated with each other (Spearman’s rank correlation coefficient *r_s_* = 0.76, *p* < 0.001) but not with the expression of *CTDSPL.* However, a significant correlation was found between the expression of *CTDSPL* and *RB1* (*r_s_* = 0.38, *p* < 0.001). This result did not agree with TCGA data, according to which the *CTDSPL* and *CTDSP2* genes are well co-expressed.

### 2.3. CTDSP1 and CTDSPL Exert Tumor Suppressive Activity In Vitro

Caki-1 is a metastatic kidney carcinoma cell line, which was established in 1971 from the cutaneous metastasis of the kidney carcinoma of a 49-year-old man. When transplanted, these cells form tumors of clear cell histology in nude mice. This line is a useful preclinical model that is very widely used in cancer research [24]. Despite a long stay in culture, Caki-1 cells are able to form structures that resemble kidney tissue in their morphology, physiology, and biochemistry [25]. Its morphologic phenotype most closely matches the morphology of metastatic ccRCC.

The cells were transfected with expression constructs containing the coding regions of the SCP phosphatase genes. We obtained three variants of clones of Caki-1 cells expressing *CTDSP1*, *CTDSP2*, and *CTDSPL*. In cells with a green fluorescent signal, the growth rate (number of cell doublings per day) was measured in the clones compared with the control cell line transfected with empty vector pT2/HB. Data analysis showed that the exogenous expression of *CTDSP1* and *CTDSPL* genes inhibited the growth of Caki-1 cells in vitro (Figure 2A–C). Statistically significant differences in the number of transfected and non-transfected cells were observed at 96 h after transfection for *CTDSP1* and *CTDSPL*.

### 2.4. Survival Analysis and Expression Analysis of SCP Subfamily Genes and RB1 in ccA and ccB Subtypes

Using the UALCAN web portal (http://ualcan.path.uab.edu, accessed on 19 August 2023), which allows analyzing gene expression based on TCGA RNA-Seq data [26], we estimated expression changes of *CTDSP1*, *CTDSP2*, *CTDSPL*, and *RB1* in two major molecular subtypes of ccRCC, namely ccA (clear cell type A) and ccB (clear cell type B) (Figure 3A–D). The expression of *CTDSP1* and *RB1* genes is predominantly increased in ccA (no statistically significant changes in ccB), while the expression of *CTDSP2* is decreased only in the ccB subtype and is almost intact in ccA. The *CTDSPL* expression is reduced in both subtypes (*p* < 0.001), but a more significant decrease is noticed for ccB. In addition, using the GEPIA web server, which provides interactive patient survival analysis based on TCGA data [27], we found that low expression of *CTDSP1*, *CTDSP2*, *CTDSPL*, and *RB1* was associated with poorer overall survival in ccRCC (*p* < 0.001, log-rank test) (Figure 3E–H). Moreover, for *CTDSPL*, the difference in overall survival rates is the most significant. Disease-free survival analysis also showed an association of lower *CTDSPL* expression with a shorter time to tumor recurrence (*p* < 0.001, log-rank test). We also found that the increased expression of the predicted regulatory miRNAs for *CTDSP1/2/L* (mir-183, mir-15a, and mir-18a; see Appendix A), is associated with worse prognosis in ccRCC (*p* < 0.01 for mir-183 and mir-18a; *p* < 0.05 for mir-15a, log-rank test; Appendix A). At the same time, the expression of mir-15a is more strongly decreased in the ccA subtype as compared to ccB (Appendix A).

## 3. Discussion

The members of the subfamily of small C-terminal domain phosphatases (CTDSP, or SCP) carry out specific dephosphorylation of serine and threonine residues in their target proteins, which are involved in a variety of biological processes. Most of these processes are often disrupted in cancer [5,13,16,17]. The substrates of SCP phosphatases include RNA polymerase II; the key cell cycle regulator Rb; SMAD transcription modulators (Figure 4); AKT1 protein kinase, which is a regulator of the cell cycle, apoptosis, and angiogenesis; transcription factors TWIST1 and c-MYC; protein of promyelocytic leukemia (PML); and others [28]. Dysfunction or inactivation of SCP phosphatases contributes to the development of various cancers, including renal carcinoma. There has been increasing interest in SCP phosphatases owing to their tumor-suppressive or oncogenic properties as well as their participation in the development of malignant tumors of various etiology and localization. For example, the deregulation of *CTDSPL* (*SCP3*) in ccRCC may lead to the deregulation of a number of important pathways in which it is involved at the mRNA level and at the level of protein interactions (Figure 4).

In a previous study [16], Yu-Ching Lin et al. demonstrated that all three phosphatases, CTDSP1, 2, and L are capable of dephosphorylating promyelocytic leukemia protein (PML), a well-known tumor suppressor, which results in the subsequent inhibition of the mTOR/HIF pathway. This may be the leading mechanism of tumor suppressive activity exerted by SCP phosphatases in ccRCC. In ccRCC cell lines 786-O and A-498, the ectopic expression of *CTDSP1* suppressed proliferation, migration, and invasion of tumor cells in vitro and in vivo (no data regarding *CTDSP2* and *L*). In the present study, we used another cell line model, Caki-1, which corresponds to metastatic ccRCC. We revealed that *CTDSP1* and *CTDSPL*, but not *CTDSP2*, are capable to suppress Caki-1 cell proliferation.

Yu-Ching Lin et al. found *CTDSP1* and *CTDSPL* expression to be frequently downregulated in ccRCC. Their downregulation correlated with PML phosphorylation at Ser-518 as well as PML downregulation; downregulation of these phosphatases was associated with high-grade tumors [16]. In our study, a noticeable decrease of expression was observed only in the case of *CTDSPL*; the expression levels of *CTDSP1* and *CTDSP2* were correlated (*p <* 0.05) but not downregulated in ccRCC. In addition, the decrease in expression of *CTDSPL* and *RB1* was more pronounced in Stage III tumors (*r_s_* = 0.38, *p* < 0.001 for correlations with stage). The correlation of *CTDSPL* downregulation with poorer survival in ccRCC patients also suggests that this gene is associated with tumor malignancy.

Among the three SCP phosphatases, it is most probable that *CTDSPL* played a leading role in the pathogenesis of ccRCC. *CTDSPL* inactivation in kidney tumors may be mediated by mechanisms different from *CTDSP1* and *CTDSP2*. In the present work, cell growth-inhibiting activity was observed for *CTDSP1* (*p* ≤ 0.001) and *CTDSPL* (*p* ≤ 0.05). These results are fundamentally different from our recent data obtained for non-small-cell lung cancer (NSCLC), in which the *CTDSP1/2/L* genes were often inactivated and all of them demonstrated tumor suppressive activity in vitro, leading to a significant slowdown of growth and senescence of the A549 lung adenocarcinoma cell line [21]. A very frequent (84%) and highly concordant (*r_s_* = 0.53–0.62, *p* ≤ 0.01) downregulation of *CTDSP1/2/L* and *RB1* was characteristic of primary NSCLC samples.

It is possible that *CTDSPL* inactivation at the mRNA level is mediated by mechanisms different from those involving *CTDSP1* and *CTDSP2* in ccRCC. Thus, intronic enhancer hypermethylation and miRNA interference may play a role. Inactivation of *CTDSPL* may be also due to deletions of the chromosome 3p locus, as indicated by previous studies (*CTDSPL* is located on chromosome 3p, which undergoes frequent deletions and hypermethylation in ccRCC) [29]. However, according to TCGA data, somatic deletions only in the *CTDSP1* gene are a frequent (up to 20% of samples of some cancers) driver event (GISTIC algorithm) in a variety of cancers, including kidney, lung, bladder, ovarian, breast, mesothelioma and other tumors. Somatic deletions in CTDSP1 are considered a likely driver event only in ccRCC and mesothelioma, and there is no evidence for deletions in CTDSP2 to be a driver event (Broad Institute, TCGA Copy Number Portal, https://portals.broadinstitute.org/tcga/gistic/browseGisticByGene, accessed on 19 August 2023).

ccRCC is a very heterogeneous type of cancer and the presence of at least two major molecular subtypes, ccA and ccB, has been previously shown [30,31]. According to the results derived from the UALCAN web resource, ccB is characterized by a lower expression of *CTDSP1*, *CTDSP2*, and *RB1*, as compared to ccA (Figure 3A–D). This is consistent with the data that patients with the ccA subtype have significantly better survival rates than those with ccB [30]. Moreover, the association of decreased expression of all four genes, *CTDSP1/2/L* and *RB1* with worse survival in ccRCC was observed according to TCGA data (GEPIA; Figure 3E–H). Despite this, a noticeable decrease in expression was found in primary tumor samples only for the *CTDSPL* gene in our sampling (Figure 1). According to these data, the expression of *CTDSP1* and *RB1* is predominantly increased (Figure 1), which indicates that the ccA subtype prevails in our sampling. Generally, it is known that ccB is characterized by increased expression of genes associated with the cell cycle and the epithelial-mesenchymal transition (EMT), while ccA is characterized by increased expression of genes associated with hypoxia and angiogenesis [32,33]. Therefore, given the ability of SCP phosphatases to block EMT and their involvement in the regulation of Rb and PML activity, the decreased expression of *CTDSP1* and *CTDSP2* in the ccB subtype in comparison to the ccA subtype looks consistent.

MicroRNAs can regulate gene expression and are one of the additional factors affecting it, like mRNA expression. In the case of RCC, it is known that micro RNAs can determine tumor progression and serve as a diagnostic and prognostic tool [34]. It has been shown that microRNAs may possess oncogenic or oncosuppressor activities [35].

The increased expression of *CTDSP1* in the ccA subtype may be due to the decreased expression of mir-183 in ccA as compared to ccB (Appendix A). Further study of the heterogeneity of ccRCC and the genetic characteristics of its subtypes is necessary for a more accurate diagnosis and the development of more effective targeted therapy, taking into account the genetic characteristics of the particular ccRCC subtypes.

It was found that the expression of *RB1* is also associated with poor survival in ccRCC patients (GEPIA, Figure 3H). Interestingly, the expression of *CCND1* is increased in ccRCC; this gene encodes cyclin D, which in complex with cyclin-dependent kinases Cdk4 and Cdk6 phosphorylates Rb during the cell cycle [36], (Appendix A). As in the case of Rb, a decrease rather than an increase in the expression of CCND1 is associated with poor survival (Figure 3H). The increase in expression of *RB1* at the early stages of ccRCC appears to be associated with its anti-apoptotic properties, regardless of its ability to block cell proliferation [37].

## 4. Materials and Methods

### 4.1. Tissue Specimens, Clinical and Pathological Characteristics

The present study included ccRCC tumors and adjacent histologically normal tissue specimens taken from 52 patients after surgical resection. Prior to the operation, the patients did not receive radiation or chemotherapy. Tumor specimens were characterized according to the international TNM tumor classification system [3] in the Blokhin National Medical Research Center of Oncology of the Ministry of Health of the Russian Federation. The clinical diagnosis was also confirmed by pathological examination of specimens at the Department of Tumor Pathologic Anatomy, Research Institute for Clinical Oncology (Moscow, Russia). Voluntary written consent was obtained from each patient to participate in the study. The use of biological specimens in the present study was within the scope of the Declaration of Helsinki and was also approved by the Ethical Committee of Blokhin National Medical Research Center of Oncology. The general clinical characterization of the specimens is presented in Appendix A.

### 4.2. Cell Culture

In this study, we used the Caki-1 cell line, a popular model of metastatic kidney carcinoma [24]. Caki-1 cells were kindly provided by Dr. Maria Kost-Alimova (Karolinska Institute, Stockholm, Sweden). Cells were cultured in DMEM medium (PanEco, Moscow, Russia) with the addition of 10% fetal bovine serum (FBS; HyClone, Logan, UT, USA), 2 mM L-glutamine (PanEco, Moscow, Russia), and 40 μg/mL gentamycin (PanEco, Moscow, Russia) in the atmosphere containing 5% CO_2_ at 37 °C.

### 4.3. Cell Transfection and Plasmids

For transfection of Caki-1 cells, genetic constructs pT2/HB-CTDSP1-2A-EGFP, pT2/HB-CTDSP2-2A-EGFP, and pT2/HB-CTDSPL-2A-EGFP were obtained according to the previously developed method [21]. Protein-coding sequences of *CTDSP1*, *2*, and *L* genes and enhanced green fluorescent protein (*EGFP*) gene were merged through the T2A linker and then cloned with the pT2/HB vector. This design allowed us to detect the expression of transfected constructs containing *CTDSP1*, *2*, and *L* genes by measuring the level of EGFP fluorescence. The vectors were introduced into Caki-1 cells by electroporation using a X-Cell Electroporation System (Bio-Rad, Hercules, CA, USA) and the Sleeping Beauty transposon system. Cell line Caki-1 transfected with the empty vector pT2/HB was used as a control. The pCMV (CAT) T7-SB100 vector was kindly provided by Dr. Zsuzsanna Izsvak (Max Delbrück Center for Molecular Medicine in the Helmholtz Association, Berlin, Germany; Addgene plasmid # 34879) and pT2/HB was kindly provided by Dr. Perry Hackett (College of Biological Sciences, University of Minnesota, Minneapolis, MN, USA; Addgene plasmid # 26557).

The efficiency of transfection for *CTDSP1*, *CTDSP2*, and *CTDSPL* was 10.3%, 22.7%, and 12.5%, respectively. The transfection efficiency of the control plasmid pTagRFP (Evrogen, Moscow, Russia; cat # FP141), encoding for the TagRFP fluorescent protein, was 19.5%. Cell sorting was performed 48 h after the transfection. The sorting was conducted using an S3 cell sorter (Bio-Rad, Hercules, CA, USA). The transfected cells were sorted for the presence of a green fluorescent signal indicating the expression of phosphatase gene inserts. After sorting, cells were plated into Costar 24-well plates (Corning, NY, USA) at 10,000 cells per well. The growth rate (number of cell doublings per day) of cells with the green fluorescent signal was determined in clones and compared to a control cell line. Cells were counted at 48, 72, and 96 h after seeding using a Diaphot inverted phase contrast microscope (Nikon, Tokyo, Japan) (4 × lens) and a Nikon D5000 camera.

### 4.4. Bioinformatics Analysis

The differential expression of *CTDSP1/2/L* and *RB1* and survival analysis of TCGA data was performed using GEPIA (Gene Expression Profiling Interactive Analysis) web server [27]. Survival curves were created using the Kaplan-Meier method with a cutoff along the median. Differences in survival curves were compared using a log-rank test, with *p*-values less than 0.05 considered statistically significant. To evaluate the differential expression between two major ccRCC subtypes (ccA and ccB), we used the UALCAN web resource, which is also based on TCGA data [26].

Also, to estimate the differential expression of *CTDSP1/2/L* and *RB1* genes in ccRCC (TCGA data) and reveal possible mechanisms of their expression regulation, we used our previously developed CrossHub tool (https://sourceforge.net/projects/crosshub/, accessed on 19 August 2023) [38]. TCGA RNA-Seq dataset (KIRC) included 533 tumors and 72 matched normal tissues. Additionally, we examined methylation profiling (320 tumors and 160 normal samples) and miRNA (254 and 71 samples, accordingly) ccRCC datasets in order to reveal possible mechanisms of SCP downregulation.

### 4.5. Quantitative Gene Expression Analysis with RT-PCR

Total RNA was isolated from tumor and normal kidney tissue samples according to the manufacturer’s protocol with commercial miRneasy Mini Kit (Qiagen, Germantown, MD, USA). The quantity and quality of RNA were evaluated using the Nanodrop-ND 1000 UV-Vis Spectrophotometer (Thermo Fisher, Waltham, MA, USA). For the reverse transcription reaction, we used the Reverse Transcription System kit (Promega, Madison, WI, USA). Real-time quantitative PCR was performed using the TaqMan^®^ Gene Expression Assay (Thermo Fisher Scientific, Waltham, MA, USA) for each gene. The relative level of mRNA of each gene was calculated using the ∆∆Ct method [24] with the *GAPDH* and *RPN1* genes as endogenous control.

### 4.6. RT-PCR Statistical Analysis

Statistical significance of differences in gene expression between tumor and matched normal samples were determined using the paired Wilcoxon signed-rank test. Statistical significance of observed changes in gene expression associated with different clinicopathological characteristics of ccRCC patients was determined using the non-paired Mann-Whitney U test. Changes were considered statistically significant at *p*-values ≤ 0.05. To estimate the possible co-expression of the studied genes, Spearman’s rank-order correlation was used.

## 5. Conclusions

Our data confirmed that *CTDSP1* and *CTDSPL* (but not *CTDSP2*) exerts tumor suppressive activity in ccRCC. However, which phosphatases exhibit tumor-suppressive activity may strongly depend on the cell line selected for the assay.

Together with differential expression data (RT-qPCR, TCGA) as well as TCGA survival data, this indicates a significant role of CTDSP/SCP family phosphatases in cancer development. However, the separation of ccRCC into molecular subtypes may be of great importance, on which the role of phosphatases in cancer development may also strongly depend. The possible role of some miRNAs in the regulation of phosphatase gene expression is also noteworthy, but this issue requires further investigation.

Understanding the role of SCP phosphatases in oncogenesis continues to be of great interest, given their known functions and their effects on the properties of tumor cells. The elucidation of the biological roles of SCP phosphatases and the design of methods of their regulation represent a vast area of further research and could lead to the development of new approaches for the treatment of kidney cancer.

## Figures and Tables

**Figure 1 ijms-24-12986-f001:**
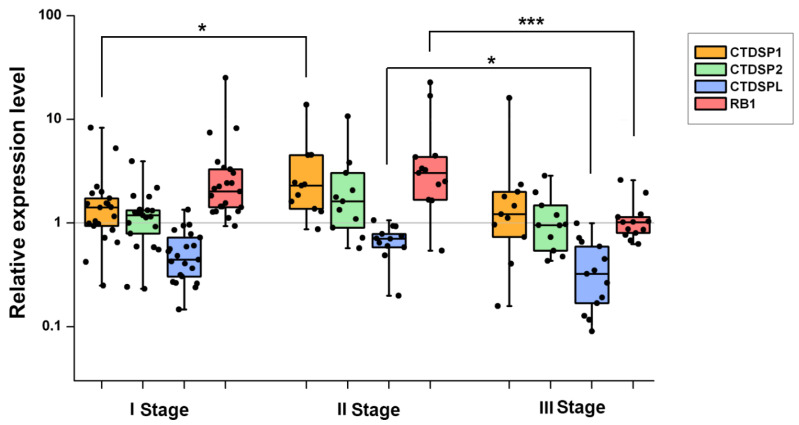
The results of quantitative RT-PCR analysis of *CTDSP1/2/L* and *RB1* relative expression in ccRCC. Boxes represent interquartile ranges (25th–75th percentiles); horizontal lines inside boxes represent medians. Vertical whiskers extend to the maximum and minimum values. * *p* ≤ 0.05; *** *p* ≤ 0.001 (Mann–Whitney U-test).

**Figure 2 ijms-24-12986-f002:**
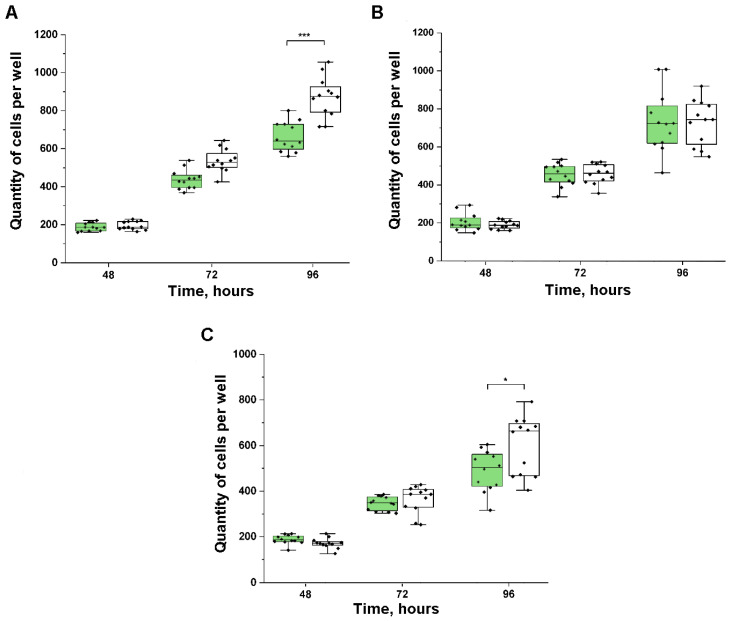
Growth rate of CaKi-1 cells after transfection. The numbers of cells transfected with plasmids containing *CTDSP1* (**A**), *CTDSP2* (**B**), and *CTDSPL* (**C**) after 48, 72, and 96 h. The rectangle represents the interquartile range (25th–75th quartile). The statistical significance of the differences among the growth rates of Caki-1 cells transfected with *CTDSP1/2/L* (green) and cells transfected with an empty vector (white) was determined by the nonparametric Mann–Whitney test. * *p* ≤ 0.05; *** *p* ≤ 0.001.

**Figure 3 ijms-24-12986-f003:**
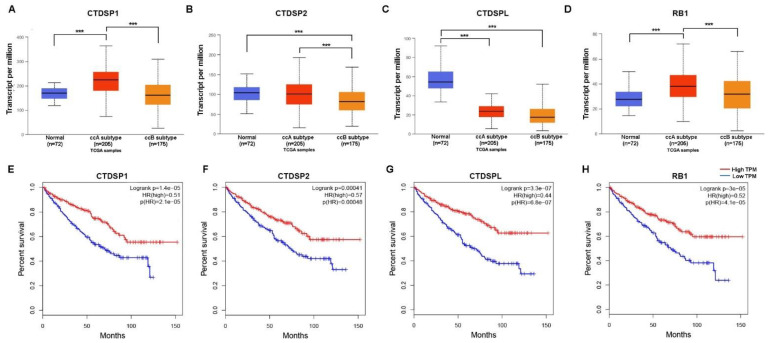
Analysis of *CTDSP1*, *CTDSP2*, *CTDSPL*, and *RB1* expression in ccA and ccB subtypes in TCGA ccRCC (KIRC dataset) samples (**A**–**D**) and survival analysis in ccRCC (**E**–**H**). (**A**–**D**) Boxplots show the median, first, and third quartiles (25th and 75th percentiles); minimum and maximum sample values; and outliers. *** *p* ≤ 0.001. Plots (**E**–**H**) show Kaplan-Meier curves for overall survival (TCGA data) with median cut-off. The log-rank test was used to assess the significance of the differences between groups with high and low transcripts per million (TPM) and the Hazards Ratio values were calculated for each gene.

**Figure 4 ijms-24-12986-f004:**
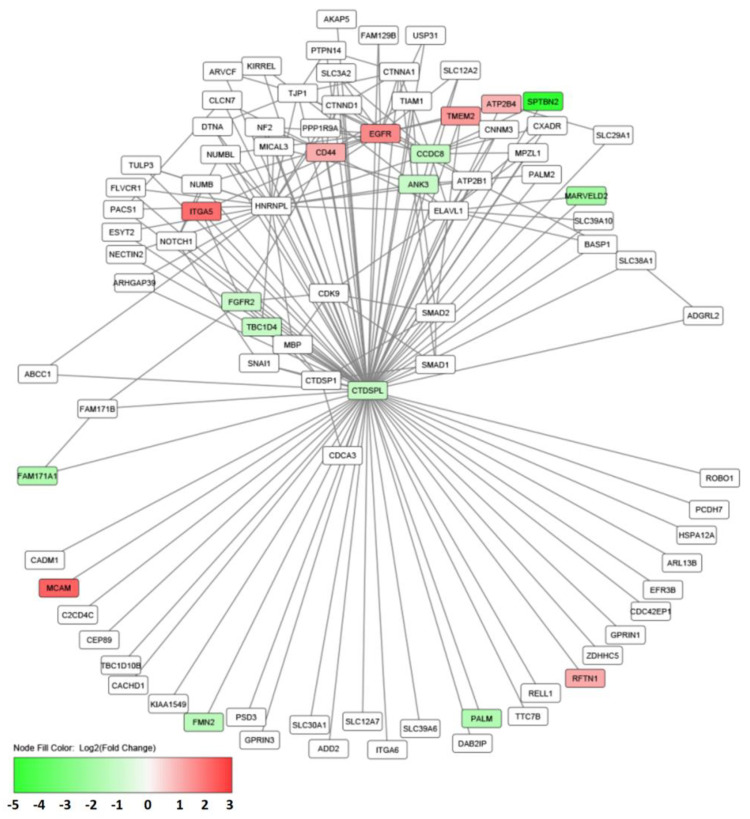
Protein-protein interactions networks (including the data from high-throughput screens) for CTDSPL taking into account differentially expressed genes in ccRCC. The genes with decreased and increased expression are marked with a gradient from green to red (color scale represents log2 of expression level fold change, tumor versus normal). The network was inferred using the GPS-Prot (BioGrid data). Differentially expressed genes in ccRCC (TCGA; KIRC dataset) were derived with ANOVA algorithm using GEPIA2.

**Table 1 ijms-24-12986-t001:** Summary of quantitative PCR data for *CTDSPL*, *CTDSP1*, *CTDSP2*, and *RB1* genes in ccRCC.

	*CTDSP1*	*CTDSP2*	*CTDSPL*	*RB1*
Average mRNA level fold change, n-fold	2.3↑ *	1.5↑ *	2.1↓ *	3.3↑ *
frequency of decrease, %	10	4	50	2
average mRNA level decrease, n-fold	3.1	2.8	3.4	4
frequency of increase, %	31	17	–	50
average mRNA level increase, n-fold	5	3.8	–	3.3

Note: ↑, increase; ↓, decrease; * *p* < 0.05 for each value.

## Data Availability

The analyzed data sets generated during the study are available from the corresponding author upon reasonable request.

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
