# Peer review of "Tumor Suppressor Properties of Small C-Terminal Domain Phosphatases in Clear Cell Renal Cell Carcinoma"

_ijms, 2023, doi:10.3390/ijms241612986_

Round 1

Reviewer 1 Report

Krasnov and colleagues have investigated 3 SCP phosphatases (CTDSP1,-2 and -L) for their potential tumor suppressor role in ccRCC. They saw differential  expression of these genes and several targets including RB1 in the TCGA ccRCC cohort,  and also in a validation cohort of 52 ccRCC patients. They showed that transfection of CTDSP1 and -L into Caki-1 RCC cells inhibited tumor growth in vitro. They found that lower expression of the 3 SCP phosphatases and RB1 is associated with poor survival in the TCGA ccRCC dataset and that reduced expression of CTDSP2 in particular is associated with an aggressive ccRCC phenotype, type ccB. They conclude that CTDSP1 and -L are tumor suppressors whose reduced expression is associated with poor survival in ccRCC.

Major point to consider:

The paper is well written and builds on work of Yu-Ching Li et al that is cited here pointing to a role of the SCP phosphatases as tumor suppressor genes in ccRCC. Lin showed that the SCPs dephosphorylate tumor suppressor PML at S518 thereby blocking PML ubiquitination and degradation. These authors should determine if the tumor growth inhibition observed by transfection of CTDSP1 and -L into Caki-1 was due to a similar mechanism. The authors should pursue this question by evaluating levels of PML S518 phosphorylation in Caki-1 cells +/- transfected SCPs. They could further pursue this line of investigation by performing in vivo experiments, generating Caki-1 mouse xenografts and determining if stable transfection of SCPs in the cells inhibited tumorigenesis in mice. These experiments would provide important information related to the mechanism of tumor suppression provided by SCPs in the setting of ccRCC.

Minor points to address:

1) What is known about the VHL status of the 52 ccRCC cohort? Please add this information to Table S1.

2) What is known about the chromosome 3p loss in the 52 ccRCC tumors? If one chr3p arm is lost in the tumor it will have only one copy of CTDSPL, so will be haploinsufficient for this tumor suppressor. They state that CTDSPL expression is reduced by 50% in this dataset. This could simply be reflecting 3p loss which is known to occur in the majority of ccRCC.

3) Did the authors evaluate 3p loss in the TCGA ccRCC dataset ? This information should be included.

3) Lin et al observed reduced expression of SCP1 and SCP3 in 36 ccRCC, whereas these authors only saw reduced expression in CTDSPL/SCP3 in TCGA ccRCC.  Can the authors speculate why these discrepancies may have occurred? Should include in discussion.

Author Response

â„–1 Comments and Suggestions for Authors

Krasnov and colleagues have investigated 3 SCP phosphatases (CTDSP1,-2 and -L) for their potential tumor suppressor role in ccRCC. They saw differential expression of these genes and several targets including RB1 in the TCGA ccRCC cohort,  and also in a validation cohort of 52 ccRCC patients. They showed that transfection of CTDSP1 and -L into Caki-1 RCC cells inhibited tumor growth in vitro. They found that lower expression of the 3 SCP phosphatases and RB1 is associated with poor survival in the TCGA ccRCC dataset and that reduced expression of CTDSP2 in particular is associated with an aggressive ccRCC phenotype, type ccB. They conclude that CTDSP1 and -L are tumor suppressors whose reduced expression is associated with poor survival in ccRCC.

Major point to consider:

The paper is well written and builds on work of Yu-Ching Li et al that is cited here pointing to a role of the SCP phosphatases as tumor suppressor genes in ccRCC. Lin showed that the SCPs dephosphorylate tumor suppressor PML at S518 thereby blocking PML ubiquitination and degradation. These authors should determine if the tumor growth inhibition observed by transfection of CTDSP1 and -L into Caki-1 was due to a similar mechanism. The authors should pursue this question by evaluating levels of PML S518 phosphorylation in Caki-1 cells +/- transfected SCPs. They could further pursue this line of investigation by performing in vivo experiments, generating Caki-1 mouse xenografts and determining if stable transfection of SCPs in the cells inhibited tumorigenesis in mice. These experiments would provide important information related to the mechanism of tumor suppression provided by SCPs in the setting of ccRCC.

Thank you very much for a very qualified review of the paper.

Our paper is mainly devoted to investigating the role of SCP phosphatases in relation to ccRCC. To answer the question whether the suppressor role of these phosphatases is specific to ccRCC tumor. Your proposed study goes far beyond the scope of this goal. Certainly, in the future it is reasonable to conduct a corresponding study involving several types of cultured ccRCC cell lines, as well as animals experiments about which you wrote.

Minor points to address:

1) What is known about the VHL status of the 52 ccRCC cohort? Please add this information to Table S1.

VHL status of the 52 ccRCC has not been investigated.

2) What is known about the chromosome 3p loss in the 52 ccRCC tumors? If one chr3p arm is lost in the tumor it will have only one copy of CTDSPL, so will be haploinsufficient for this tumor suppressor. They state that CTDSPL expression is reduced by 50% in this dataset. This could simply be reflecting 3p loss which is known to occur in the majority of ccRCC.

Unfortunately, genetic analysis of the tumors was not performed.

3) Did the authors evaluate 3p loss in the TCGA ccRCC dataset ? This information should be included.

The genes of the phosphatases we studied are located on different human chromosomes (CTDSP1 – 2q35, CTDSP2 – 12q14.1, and CTDSPL – 3p22.2), so we did not study the loss of the small arm of chromosome 3 in this study.

3) Lin et al observed reduced expression of SCP1 and SCP3 in 36 ccRCC, whereas these authors only saw reduced expression in CTDSPL/SCP3 in TCGA ccRCC.  Can the authors speculate why these discrepancies may have occurred? Should include in discussion.

In the Discussion section (lines 226-243), we consider possible reasons for the discrepancy between our data and those obtained in the study:

Lin, Y.C.; Lu, L.T.; Chen, H.Y.; Duan, X.; Lin, X.; Feng, X.H. et al. SCP phosphatases suppress renal cell carcinoma by stabilizing PML and inhibiting mTOR/HIF signaling. Cancer Res. 2014, 74, 6935-6946.

Unfortunately, we are unfamiliar with the work you write about. Please give us a specific reference.

In general, discrepancies may stem from the unusually high heterogeneity of ccRCC, sampling specificities, differences in methodological approaches, etc.

Reviewer 2 Report

The authors should be congratulated for their work and for addressing an important topic. However, some points warrant mentions:

MAJOR COMMENTS:

1.     In the “Introduction” section, line 62, the authors are invited to better explain why they choose to repeat the study by reporting its limitations. In my opinion, following this approach, the authors could also better explain the rationale behind their study. The aim of the study is not clearly expressed in the “Introduction” section, indeed.

MINOR COMMENTS:

1.     I suggest to avoid abbreviations in the Title, as well as defining SCP in the “Abstract”.

2.     in the “Introduction” section, lines 41-42, the sentence is too short to give the readers an overview of the ccRCC. In this context, I suggest that the authors expand this sentence according to EAU/AUA guidelines and also include an overview of therapies and adverse effects as in PMID: 37353178.

3.     in the “Introduction” section, the authors are invited to define SCP at the first use.

4.     The “S1” table is totally not understandable, and I can’t find any scientifical meaning. The authors are invited to report results by means and standard deviations of median and interquartile range. Moreover, I suggest to eliminate TNM and reporting only stages making this table more readable.

5.     Also, the authors are invited to define all the acronymous in the Tables and Supplementary tables.

6.     In the “Discussion” section, line 272 “Most of these processes are often disrupted in cancer” needs a citation.

7.     In the “Discussion” section, line 315, ccRCC has been defined yet, so only the short version should be used. The authors are invited to check the manuscript for similar errors.

8.     In the “Discussion” section, the authors should explain why they tested miRNA expression. In this context, a valid review is given by PMID: 37446024.

English form should be revised.

Author Response

â„–2. Comments and Suggestions for Authors

The authors should be congratulated for their work and for addressing an important topic. However, some points warrant mentions:

We thank the reviewer for the work done to improve the quality of the manuscript.

MAJOR COMMENTS:

  1. In the “Introduction” section, line 62, the authors are invited to better explain why they choose to repeat the study by reporting its limitations. In my opinion, following this approach, the authors could also better explain the rationale behind their study. The aim of the study is not clearly expressed in the “Introduction” section, indeed.

we've added lines to the introduction, lines 78-80

MINOR COMMENTS:

  1. I suggest to avoid abbreviations in the Title, as well as defining SCP in the “Abstract”.

We've done it.

  1. in the “Introduction” section, lines 41-42, the sentence is too short to give the readers an overview of the ccRCC. In this context, I suggest that the authors expand this sentence according to EAU/AUA guidelines and also include an overview of therapies and adverse effects as in PMID: 37353178.

We have inserted additional information about ccRCC: lines 42-48.

The outcomes of ccRCC are the worst among all genitourinary tumors. Renal cell carcinoma represents around 3% of all cancers, with the highest incidence occurring in Western countries. RCC is the 13th leading cause of cancer deaths worldwide.

Capitanio, U.; Bensalah, K.; Bex, A.; Boorjian, S.A.; Bray, F.; Coleman, J.; Gore, J.L.; Sun, M.; Wood, C.; Russo, P. Epidemiology of Renal Cell Carcinoma. Eur Urol. 2019, 75, 74-84. doi: 10.1016/j.eururo.2018.08.036.

In general, ccRCC is well circumscribed and the capsule is usually absent. Loss of chromosome 3p and mutation of the von Hippel-Lindau (VHL) gene located at chromosome 3p25 are common. Loss of function of the von Hippel-Lindau protein contributes to tumor initiation, progression, and metastasis. Additional ccRCC tumor suppressor genes (UTX, JARID1C, SETD2, PBRM1, BAP1) are located at the 3p locus [23].

WHO Classification of Tumours Editorial Board. Urinary and Male Genital Tumours. In: WHO Classification of Tumours, 5th Edition, Volume 8. 2022. ISBN 978-92-832-4512-4.

  1. in the “Introduction” section, the authors are invited to define SCP at the first use.

We've done it.

  1. The “S1” table is totally not understandable, and I can’t find any scientifical meaning. The authors are invited to report results by means and standard deviations of median and interquartile range. Moreover, I suggest to eliminate TNM and reporting only stages making this table more readable.

We have redesigned table S1. It's now called the S5. We removed the TNM information and added the distribution of patients by age and ccRCC stage.

  1. Also, the authors are invited to define all the acronymous in the Tables and Supplementary tables.

We've done it.

  1. In the “Discussion” section, line 272 “Most of these processes are often disrupted in cancer” needs a citation.

We've done it.

  1. In the “Discussion” section, line 315, ccRCC has been defined yet, so only the short version should be used. The authors are invited to check the manuscript for similar errors.

We've done it.

  1. In the “Discussion” section, the authors should explain why they tested miRNA expression. In this context, a valid review is given by PMID: 37446024.

We have inserted the relevant fragment, lines 277-280.

Comments on the Quality of English Language

English form should be revised.

The article was read and edited by an experienced American scientist.

Round 2

Reviewer 1 Report

The authors indicate that the experimental suggestions of this reviewer are beyond the scope of this paper. They did not evaluate VHL status or 3p loss in their ccRCC cohort and do not plan to do so. In this case, the comments of this reviewer have been answered and will not require additional revision.

Reviewer 2 Report

Thank you for addressing all my comments, the manuscript has reached higher scientific levels. Well done.